# Pan-Genotype Pre-Exposure Prophylaxis (PrEP) Allows Transplantation of HCV-Positive Donor Kidneys to Negative Transplant Recipients

**DOI:** 10.3390/jcm10010089

**Published:** 2020-12-29

**Authors:** Michael Duerr, Lutz Liefeldt, Frank Friedersdorff, Mira Choi, Robert Öllinger, Jörg Hofmann, Klemens Budde, Eva Schrezenmeier, Fabian Halleck

**Affiliations:** 1Department of Nephrology and Intensive Medical Care, Charité—Universitätsmedizin Berlin, Corporate Member of Freie Universität Berlin, Humboldt-Universität zu Berlin, and Berlin Institute of Health, Charitéplatz 1, 10117 Berlin, Germany; lutz.liefeldt@charite.de (L.L.); mira.choi@charite.de (M.C.); klemens.budde@charite.de (K.B.); eva-vanessa.schrezenmeier@charite.de (E.S.); fabian.halleck@charite.de (F.H.); 2Department of Urology, Charité—Universitätsmedizin Berlin, Corporate Member of Freie Universität Berlin, Humboldt-Universität zu Berlin, and Berlin Institute of Health, Charitéplatz 1, 10117 Berlin, Germany; frank.friedersdorff@charite.de; 3Department of Surgery, Campus Charité-Mitte and Campus Virchow-Klinikum, Charité—Universitätsmedizin Berlin, Corporate Member of Freie Universität Berlin, Humboldt-Universität zu Berlin, and Berlin Institute of Health, Charitéplatz 1, 10117 Berlin, Germany; robert.oellinger@charite.de; 4Institute of Medical Virology, Charité—Universitätsmedizin Berlin, and Labor Berlin Charité-Vivantes GmbH, 10117 Berlin, Germany; joerg.hofmann@charite.de; 5Berlin institute of Health (BIH), Anna-Louisa-Karsch-Str. 2, 10178 Berlin, Germany

**Keywords:** kidney transplantation, hepatitis C, virus transmission

## Abstract

Transplant candidates are facing incremental mortality risks on the waiting list. Here, we report a novel strategy to expand the donor pool by including hepatitis C seropositive (HCV+) donors. We investigated a pre-exposure prophylactic (PrEP) treatment with direct-acting antivirals (DAA) to allow transplantation for HCV seronegative (HCV−) kidney transplant recipients (KTR) with the aim to prevent HCV infection post transplantation. In this prospective trial, a pan-genotypic PrEP with daclatasvir and sofosbuvir once daily for 12 week was administered at transplantation. The primary endpoint sustained virological negativity (SVN) 12 weeks after the end of PrEP. Seven patients received a transplantation from four HCV+ donors. Accumulated waiting time was 70 ± 31.3 months already. Of note, study subjects underwent transplantation 24.7 ± 16.1 days after given consent. All KTR developed excellent graft function without any rejection episodes. One patient died with a functioning graft due to sepsis 13 months after transplantation. PrEP demonstrated efficacy with no signs of HCV transmission with excellent tolerability. Two out of four HCV+ donors were viremic at the time of explantation. Interestingly, KTR developed HCV antibodies also from non viramic donors. The acceptance of HCV+ donor was safe and reduced waiting time under the protection of PrEP DAA in kidney transplantation.

## 1. Introduction

Because of worldwide organ shortage, patients with kidney failure with replacement therapy (KFRT) are facing long waiting times before receiving kidney transplantation. Due to low donor rates in Germany, adult patients (18–65 years) have to wait for around 8 years before they get a transplant, while comorbidities and deaths on the waiting list are rising [1]. Therefore, novel strategies with the goal to expand the donor pool are desperately needed. It has been shown that shorter waiting times of HCV-seropositive (HCV+) donors to HCV+ recipients provide better death-censored graft and patient survival compared to HCV seronegative (HCV-) recipients waiting for an HCV- donor [2,3,4].

Since the introduction of novel direct-acting antiviral agents (DAA), HCV infection is curable in patients after renal transplantation with excellent efficacy and tolerability [5,6]. Currently, HCV+ kidney donor organs are only allocated to HCV+ kidney transplant recipients—a small minority of KFRT patients on the waiting list. Between 2017 and 2019, out of *n* = 65 HCV+ donors, *n* = 128 HCV+ kidneys were offered, of which *n* = 86 were transplanted over this period. Still, there is a 32% discard rate of HCV+ donor kidneys. Interestingly, only *n* = 29 transplantations from HCV+ donors in contrast to *n* = 3240 kidney transplantations from non-HCV donors took place on average per year.

Transplanting Hepatitis C Kidneys Into Negative Kidney Recipients (THINKER)-trial successfully demonstrated transplanting HCV+ viremic donor kidneys without the development of chronic HCV infection. Although HCV transmission occurred in all transplanted recipients, post-expositional prophylaxis (PEP) therapy resulted in 100% sustained virological response (SVR) in study participants [7,8]. Utilizing of HCV+ donor organs into HCV- recipients under the umbrella of DAAs, those patients received a timely transplant. As there is convincing data on the feasibility to use HCV+ organs for HCV- transplant recipients based on case reports [9] and small pilot studies [10,11], evidence that emerged from controlled prospective trials is needed.

Therefore, we developed a prospective study to transplant HCV+ kidney organs into HCV- recipients to show efficacy and safety by preventing HCV transmission with a pan-genotypic pre-expositional prophylaxis (PrEP) with daclatasvir (DCV) and sofosbuvir (SOF).

## 2. Methods

The study was conducted according to the Declaration of Helsinki, the International Conference on Harmonization and Good Clinical Practice guidelines. Potential trial participants agreed to participate in the study by providing written informed consent after approval by German health authorities and an independent ethics committee (15/0446EK15; 4040892). On 15 December 2017, this prospective, single-center, open-label trial (Eudra-CT number: 2016-003088-21) started in our center. Because the manufacturer terminated drug production of daclatasvir, the ongoing study was terminated before enrolling all planned patients.

### 2.1. Study Design

In close cooperation with our ethics committee, first we developed a clinical decision model in order to select those patients with expected long waiting times and highest risk for complications on the waiting list. The rationale behind this strategy was the unknown safety and efficacy of the antiviral regimen in the setting of kidney transplantation, which could result in transmission of HCV infection in a patient with a failed transplant in a worst-case scenario. To balance the risks, we selected only those patients who would benefit most and in whom the advantages of a timely transplant would outweigh the risks of virus transmission. Stratified on a clinical decision model to detect a target population with elevated mortality risk (Figure 1), waiting list candidates were identified to be eligible for study entry.

Eligible waiting list candidates were approached and educated (individually or in group sessions) on the potential risks and benefits of accepting a HCV+ donor organ, and rescue treatment options in case of resistant or refractory HCV virus infection. After given written informed consent, the individual status on the Eurotransplant waiting list was changed in order to receive an HCV+ organ. When allocation of an HCV+ organ, the risks and benefits of accepting this allocated HCV+ organ were discussed again in light of the potential organ offer and transplantation was only performed after given second informed consent (Figure 2). Further, the study had a staggered design (Figure 3) with an efficacy interim-analysis 4 weeks after transplantation and PrEP therapy initiation to demonstrate sustained virological negativity (SVN4). In phase I, only *n* = 5 patients were enrolled. If all study subjects showed SVN4, *n* = 10 more patients could be enrolled within phase II of the study. In case one out of five had lack of efficacy and did not demonstrate SVN4, the recruitment of *n* = 5 additional patients was required by protocol to complete phase I. If more than two out of 5 or two out of ten patients were HCV positive and did not show SVN4 as interim endpoint, the study had to be terminated due to a lack of efficacy. In phase II again, an interim efficacy assessment had to take place and the study had to be terminated if >10% of patients did not reach the efficacy endpoint SVN4. After phase I and II, all key data concerning safety and efficacy of all study patients were reviewed by the responsible ethic commission. A positive consent of the EC was mandatory before the next phase could be launched. Finally, in phase III, additional *n* = 24 patients were to be recruited for the analysis of the primary endpoint. Again, SVN4 and SVN 12 were to be monitored continuously. In case more than 10% of all study subjects (including patients from phase I and II) were tested HCV positive 4 weeks after end of PrEP, the study had to be terminated.

### 2.2. Key Inclusion Criteria

Subjects who are 18–49 years old were only eligible, when their projected waiting time on the waiting list is >60 months (5 years) to receive a renal transplant and display at least two clinical risk factors (diabetes mellitus, documented coronary artery disease, severe arteriosclerosis, signs of heart failure, allo-sensitized (%PRA ≥20%) OR with imminent lack of access for dialysis (either HD or PD). Subjects who are 50–59 years old are only eligible when they display at least one additional risk factor (diabetes mellitus, documented coronary artery disease, severe arteriosclerosis, signs of heart failure, allo-sensitized (PRA ≥20%) OR with imminent lack of access for dialysis (either HD or PD). Subjects who are >60 years old are eligible without any additional risk evaluation

### 2.3. Key Exclusion Criteria

Waiting list candidates with any contraindications for PrEP treatment, evidence for a medical condition associated with chronic decompensated liver disease (Child-Pugh Class B or C) were excluded from the study. Further key exclusion criteria were: history of variceal bleeding, hepatic encephalopathy or ascites, severe cardiac disease, any blood transfusions within 4 weeks, malignancies in the last 5 years, signs for active peptic ulcer disease, chronic diarrhea, or gastrointestinal malabsorption, recent (within 6 months) drug or alcohol abuse (defined by [12]), coinfection with human immunodeficiency virus (HIV) or hepatitis B virus (HBV), neutrophils ≤1500/mL, platelets ≤75.000/mL, ALT >5× upper limit of normal (ULN), direct bilirubin > 3×ULN, albumin <3.0 g/dL.

### 2.4. Treatment Regimen

After allocation and acceptance of an HCV+ kidney from deceased brain death (DBD) donor, PrEP was initiated with DCV 60 mg and SOF 400 mg orally in all participants before the transplantation procedure started. PrEP was given for 12 weeks to study participants if the HCV+ donor was viremic in the RNA testing. If donor HCV RNA was negative, PrEP was stopped four weeks after transplantation. Immunosuppression was given according to our center policy (induction therapy with basiliximab, maintenance with tacrolimus, mycophenolate, and steroids).

### 2.5. Efficacy Endpoints

The primary outcome was sustained virological negativity at week 12 after the end of treatment (SVN12). SVN was defined as undetectable HCV RNA in a study participant with either unquantifiable or undetectable HCV RNA. Secondary endpoints were SVN4 and SVN24, sustained virological negativity at week 4 and week 24, respectively.

### 2.6. Safety

We prospectively evaluated safety of treatment regimen at each visit in context of adverse events (AEs), serious AEs (SAEs), suspected unexpected severe adverse reaction (SUSAR), and drug discontinuations. Patient and graft survival including biopsy-proven rejections (BPAR) according to BANFF scores [13] were documented. Events of particular interest as infections, clinically significant changes in vital signs or laboratory parameters including new-onset or changes of proteinuria and changes of eGFR at each study visit were captured. In general, participants were encouraged to report any side effects or adverse events at each trial visit.

### 2.7. Virological Testing

HCV screening and monitoring was performed on HCV-antibody, -antigen, and -RNA level with the following commercial assays: Elecsys^®^ Anti-HCV II (Roche Diagnostics GmbH, Rotkreuz, Switzerland), recomLine HCV IgG (Mikrogen Neuried, Bavaria, Germany), Architect HCV-Ag, (Abbott Laboratories, Abbott Park, IL, USA), COBAS^®^ AmpliPrep/COBAS^®^ TaqMan^®^ HCV Quantitative Test v2.0 (Roche Diagnostics GmbH, Mannheim, Germany), VERSANT^®^ HCV Genotype 2.0 Assay Line Probe Assay (Siemens Healthcare, Erlangen, Germany).

### 2.8. Statistical Analysis

All patients are to be included for efficacy endpoints who received at least one dose of study medication (intention to treat approach). The proportion of patients with negative HCV-RNA is calculated at visits along with the 95% confidence intervals. For sample size justification, no data were available addressing treatment efficacy for this exclusive cohort. However, previous studies showed a treatment success of regimes for treatment of chronic HCV infection of at least 90%. Under the assumption of 90% SVN4, there is 41% chance that at least one of the 5 patients in phase I will have HCV detectible at week 4 after end of PrEP, and there is 26% chance that 2 or more of the 10 patients will be HCV detectible at week 4 after end of treatment. Under the 90% SVN4 assumption, the likelihood of study termination at phase I is about 17% and at phase II is about 26%. A total of 39 patients have to be enrolled in the study to estimate the rate of treatment success of post-expositional regimes assuming a precision of 10%, a significance level of 5%, and a statistical power of 80%.

## 3. Results

Here, we report the results of the first seven renal transplantations from HCV+ donors into HCV- recipients at our center before the study was terminated. First, we screened our waiting list for eligible study candidates. Based on our clinical decision model, we identified *n* = 80 potentially suitable patients that first were informed by letter about the study in general. Responding patients (*n* = 35) were offered to take part in an educational session (individual or group seminar). Thereafter, written informed consent to accept a HCV+ donor organ was given by 17 patients and their waiting list status was changed to receive a HCV+ kidney.

In total, we accepted four HCV+ donors to realize seven consecutive renal transplantations into HCV- recipients. Baseline characteristics of donors and recipients are summarized in Table 1. All recipients received their first graft and had a standard immunological risk with a negative crossmatch and a cPRA <15% in class I and II antigens.

Of note, included patients were on the active waiting list for 70 ± 31 months already. Immediately after change of the waiting list status to accept HCV+ kidneys, our patients received a transplantation after only 25 ± 16 days.

We followed the remaining patients (*n* = 10) who did not receive a HCV+ donor offer within the study timeframe. Two patients got a renal transplant at median 3.5 years after the initial study session, one patient died because of sepsis, two were removed from the waiting-list because of personal reasons, a further two were not transplantable due to suspected malignancy, and only three were still listed active as transplantable.

### 3.1. Efficacy of PrEP

*Oral* PrEP therapy was initiated in all study patients 2–3 h before renal transplantation. At end of treatment (EOT), HCV RNA was undetectable in all patients.

Three out of four HCV positive donors had a history of previous or ongoing intravenous drug use. Surprisingly, two donors did not have detectable HCV RNA in blood (Table 2). Two donors were viremic with viral loads >200.000 IU/mL at the time of explantation. Both donors had HCV 2a genotype. Out of these viremic donors, we realized three transplantations.

All three patients who received a kidney from a viremic donor exhibited shortly after transplantation detectable but very low amounts of HCV RNA, most probably passively transferred from donor tissue or plasma. Beyond day 7 after kidney transplant, as all other recipients too, were RNA negative. Of note, the transplantation of a HCV seropositive kidney led to low level HCV specific antibodies in 5/7 recipients until the end of the observation period.

### 3.2. Safety Data

In general, tolerability of administered PrEP was excellent with no treatment associated adverse events. Renal function at the end of therapy until the end of follow-up was excellent with a mean creatinine of 1.3 ± 0.4 mg/dl. Liver enzymes were regualry monitored. No elevation was detected in patients 1–6 during the follow-up time. In patient 7, a mild elevation (<2-fold) was detected during admission hospital admission with sepsis (see below in long term follow up 3.4).

### 3.3. SAEs

Adverse events were closely monitored throughout the study and no graft loss or patient death was reported within the study. Patient number 1 had a dissection of the iliac artery associated with surgery that needed a stent intervention on the first postoperative day. Delayed graft function occurred in 3 of 7 patients (patient number 2, 5, 6). Two patients had one dialysis after transplantation due to elevated potassium and postoperative fluid overload (patient number 2 and 5). One patient received two hemodialysis sessions after transplantation (patient number 6) due to fluid overload. None of the patients had a biopsy proven acute rejection. One biopsy was performed showing acute tubular necrosis but no signs of rejection (patient number 2), not requiring further therapy. No further biopsies were performed in the study population. Patient number 7 had a urinary tract infection with Pseudomonas aeruginosa requiring IV antibiotic treatment and hospitalization. SAEs are summarized in Table 3.

### 3.4. Long-Term Follow-Up

The mean follow-up of patients was 2.4 (±0.53) years (Table 4). Within this period, patients 1–6 had a stable graft function without the need for any interventions (biopsy/hospitalization). Mean serum creatinine was 1.2 mg/dl (±0.22). Patient number 7 was hospitalized with recurrent urinary tract infections with pseudomonas aeruginosa and died 12 months after kidney transplant due to methicillin susceptible staphylococcus aureus (MSSA) sepsis. Kidney function declined with every episode of urinary tract infection and was 3.84 mg/dl shortly before the admission with sepsis.

## 4. Discussion

Here, we report the results of a prospective open-label single-center trial in which we evaluated the use of HCV+ donor organs for HCV- KTR using a pan-genotypic PrEP DAA regimen with DCV and SOF. Most importantly, transplantation was performed successfully in all study participants. PrEP was well tolerated and none of the KTR developed HCV infection. Taken together, our pilot study provides further evidence for the efficacy and safety of this approach, which could increase donor pool and acceptance criteria for patients in need of timely transplantation.

Goldberg et al. published the first report (THINKER trial) of ten HCV− KTR who received HCV+ donor organs in 2017 [7]. Recently, 12-month outcomes of the original THINKER trial and 6-month data on an additional ten KTR were reported [7,8]. The THINKER trial used a DAA regimen with grazoprevir and elbasvir (GZR-EBR), covering HCV genotype 1 and 4 only. Therefore, investigators installed a novel rapid HCV genotype assay before transplantation; hence 50% of potential kidney donors were excluded due to non-genotype 1 HCV [14]. Of note, all patients developed viremia after transplantation, starting of PEP DAA was necessary. Unfortunately, rapid HCV genotype testing is not always available, and targeting only genotype I narrow potential HCV donor utilization in this protocol. Besides, THINKER’s trial feasibility of transplanting HCV+ donor organs was confirmed in other small studies and case reports [9,11] under real world conditions with good results and different treatment protocols that covered all HCV genotypes.

In contrast to previous studies, we report a study using a pan-genotypic DAA PrEP in KTR to prevent HCV transmission. This approach broadens the donor pool substantially as our donors with genotype 2 would not have been suitable for treatment with e.g., THINKER trial protocol. Due to our study design, acceptance of all genotypes from HCV+ donors was doable, and time-sensitive KTR was performed without waiting for genotype testing. We started PrEP DAA shortly before the transplant procedure to avoid logistic challenges of pre-transplant HCV RNA determination to utilize HCV+ donors promptly. Other larger studies show excellent results with the pan-genotypic treatment regimen glecaprevir and pibrentasvir over eight [15] or only four weeks [16].

Treatment with DAA was performed for 12 weeks in KTR who received kidneys from HCV+ viremic donors. In seropositive but RNA negative HCV donors, treatment in KTR was shortened to 4 weeks only. The reason to give PrEP DAA also in the setting of non-viremic donors reflects a cautious approach in the context of patient safety. The chosen PrEP protocol is simple to implement and reflects a safe and effective approach to avoid unnecessary HCV viremia or acute hepatitis. Recent data suggest that even more cost-effective, ultra-short treatments for only seven [17] or four days [18] might be sufficient to avoid HCV transmission.

Notably, SOF is only approved for eGFR >30 mL/min, limiting its use in KTR directly before or immediately after kidney transplantation. However, there is growing evidence that treatment with SOF based regimens is safe in these patients [19]. Recently published data from our group and others provide further evidence that SOF does not accumulate in patients with low GFR [19,20,21,22]. In the meantime, newer combinations are available which can be administered irrespective of GFR. Molnar et al. reported 47 patients who were cured by receiving glecaprevir plus pibrentasvir of which achieved SVR12 [23].

Of note, all HCV+ donors in our study were younger (46.4 (±7.8)) than the median donor age of 50.4 years at our center, reflecting excellent transplant kidney function in all HCV+ recipients. All donors died from either drug overdose, traumatic ICB, or aspiration. The high proportion of patients dying from drug overdose requires extensive screening for HIV and hepatitis B, which otherwise could be transmitted to the recipients. For this reason, recipients of such organs have to be informed about the potential risks and give informed consent for potentially unknown risks.

We developed a cautious study design to select those patients whose benefits outweigh the potential risks. This stepwise approach could serve as a template for other experimental treatments with unclear risks. Like in our study, standardized patient education is indispensable and can encourage patients to accept HCV+ organs [24]. Our multistep informed consent process based on waiting time and perceived risks for morbidity and mortality can guide other transplant centers to select the best-qualified waitlisted patients for accepting HCV+ donor organs.

The kidney recipients from the viremic donor had very soon after the transplantation detectable HCV RNA levels below the lower limit of detection (15 IU/mL). It is conceivable that RNA was transmitted through body liquids. All RNA determinations afterwards were continuously negative, if this indicates a true prophylaxes with DAA or a very early treatment efficacy remains speculative. As HCV+ viremic kidney donors’ transmission rates occurred in previous studies in almost 100%, from our point of view, a prophylactic treatment regimen for HCV− KTR represents an optimal approach instead of postponing treatment until viremia is detectable. In addition, a pangenotypic regimen offers the possibility to start transplantation without any delay and a timely DAA prophylaxis before transplantation could result in an overall shorter treatment duration and cost efficiency, in contrast to start days after transplantation when viremia is detectable. This hypothesis should be addressed in future studies.

Interestingly, in 5 out of 7 patients, HCV-antibody reactivity was detected soon after transplantation. A humoral immune response to an acute infection, especially in the setting of a transplant-associated immunosuppression, within such a short time frame seems unlikely. Passive transmission of donor antibodies with residual body-liquids should disappear quickly due to their half-time. Those low-level antibodies could not be confirmed with less sensitive blot approaches. Taken together, very early and very low antibody levels over several months spanning time might result from the transmission of antibody-producing cells rather than following an acute HCV infection in the recipients. This is also supported by the observation, that 2 of those 4 patients received kidneys from HCV seropositive but non-viremic donors. It has been described recently, that antibody conversion correlates with viral load [25], and that anti-HCV antibodies are mainly of an IgG isotype that makes a primary immune response to donor HCV very unlikely [26].

Similar to all previous studies with DAA, treatment was well tolerated. We did not observe drug-related adverse events in our small study. The overall long-term outcome was excellent, with excellent graft function in 6/7 patients after two years. Proteinuria and liver function tests were unremarkable during follow-up in all patients. One patient died one year after transplantation due to infectious complications that were not associated with the study treatment.

The acceptance of HCV+ donor organs resulted in a drastic reduction of waiting time, suggesting that those HCV+ organs could not be allocated to other patients within Eurotransplant. During the study period, the mean waiting time to receive kidney transplantation from a deceased donor was 9.7 years (for blood group A; B; AB; and O were 8.7; 10.0; 5.9; and 11.6 years, respectively) at our center. Likely, the significant reduction of waiting time results in a survival benefit for these patients since morbidity and mortality on the waiting list are high [27]. A standardized patient education like in our study is indispensable and can encourage patients to accept HCV+ organs [24].

The small number of study patients is a limitation caused by the unavailability of daclatasvir supply due to the manufacturer’s decision to withdraw the drug from the market. However, most recently, alternative DAAs have been tested in preventing HCV transmission in HCV+ to HCV-transplants also addressing the necessary duration of treatment [16]. Our study’s strengths include the detailed prospective study design with rigorous considerations of ethical issues, safety, and the simple and successful strategy for all patients, irrespective of genotype and RNA determinations.

## 5. Conclusions

In this prospective study, PrEP with DAA for HCV-seronegative transplant recipients of HCV+ donor kidneys was safe and well-tolerated. Neither transplant-associated HCV infection in any recipients nor adverse events related to treatment regimen occurred within the study. In addition, in the longer follow up, no case of chronic donor-derived HCV infection was detectable. We were able to provide long-term data on follow-up of our cohort. As reimbursement of DAAs in this setting remains unclear, we believe cost-effectiveness should also be addressed in future trials as this strategy should markedly expand organ options and reduce mortality for HCV− KT candidates.

## Figures and Tables

**Figure 1 jcm-10-00089-f001:**
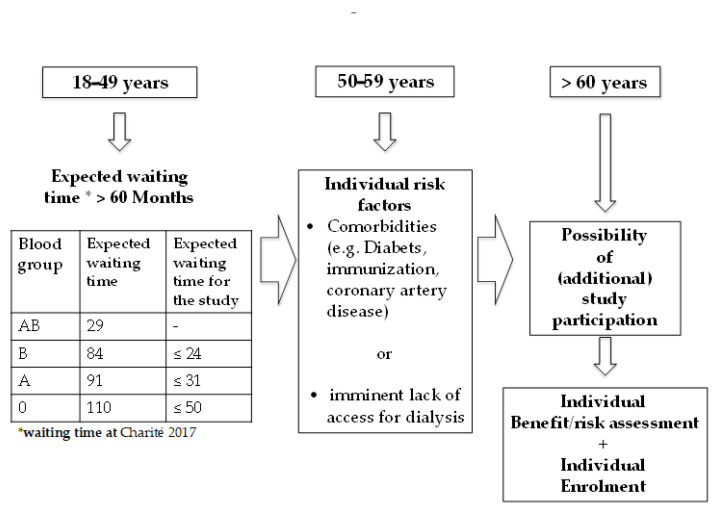
Decision criteria for participation in the study. Patients were stratified on a clinical decision model to detect a target population with elevated mortality risk.

**Figure 2 jcm-10-00089-f002:**
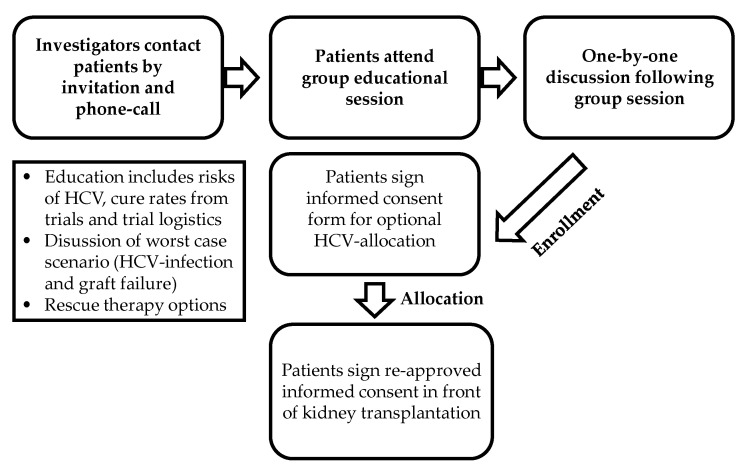
Process of informed consent. Eligible waiting list candidates were approached and educated (individually or in group sessions) on the potential risks and benefits of accepting a hepatitis C seropositive (HCV+) donor organ, the worst-case-scenario and rescue treatment options in case of resistant or refractory HCV virus infection.

**Figure 3 jcm-10-00089-f003:**
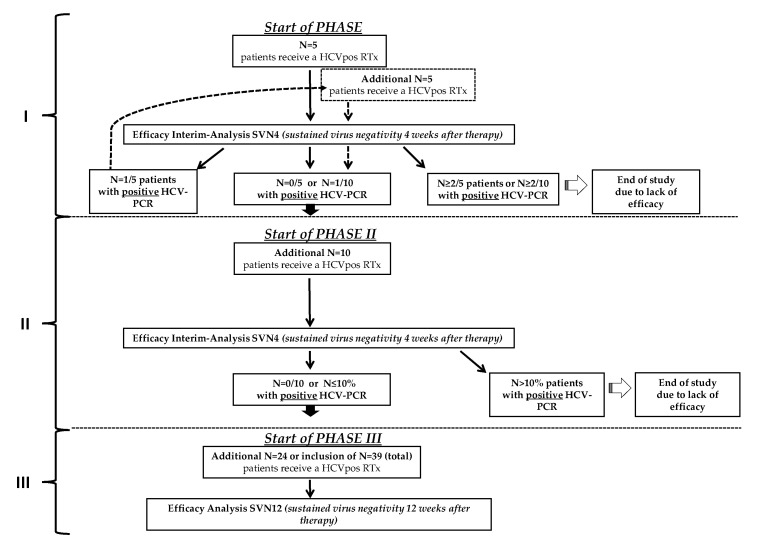
Staggered study design. The flowchart reflects the stepwise cautious enrollment of *n* = 5 patients in phase I, *n* = 10 in phase II, and an additional *n* = 24 in phase III. Within each phase, a prespecified efficacy interim analysis SVN4 (sustained virus negativity 4 weeks after therapy) was performed in each phase of the study. In phase I, study would be terminated if more than 2 out 5 or 2 out of 10 patients tested positive for HCV transmission. In all other phases, study termination was defined by a HCV transmission rate >10% in interim efficacy analysis. RTx = renal transplantation.

**Table 1 jcm-10-00089-t001:** Baseline characteristics of HCV positive donors and of HCV negative recipients.

HCV Positive Transplant Donors (*n* = 4)	
Mean age, y (SD)	46.4 (±7.8)
Male, n	4
History of IV drug abuse	3
Cause of death	
ICB (traumatic)	2
Drug overdose	1
Bolus aspiration	1
Creatinine (mg/dl) (SD)	0.6 (±0.02)
HCV negative transplant recipients (*n* = 7)	
Mean age at transplantation, y (SD)	59.4 (±8.4)
Female, *n*	3
Blood group, *n*	
AB	2
B	3
A	0
0	2
Median time on ET waiting list (SD)	24.7 (±16.1) days
At study entry (SD)	70 (±31.3) months
After study entry until transplantation (SD)	24.7 (±16.1) days
Median broad HLA mismatches (IQR)	4 (3/5)
Cause of kidney failure with replacement therapy (patient No.)	1 diabetic nephropathy 2 glomerulonephritis 3 IgA nephropathy 4 interstitial nephritis 5 unknown 6 thrombotic thrombocytopenic purpura 7 unknown

ICB = Intracranial bleeding, ET = Eurotransplant.

**Table 2 jcm-10-00089-t002:** Efficacy outcome data.

HCV Positive Kidney Donors				HCV Negative Kidney Transplant Recipients								
No.	GT	HCV Ab	HCV RNA Level, IU/mL	No.	HCV RNA Level, IU/mL						HCV Ab Status	
					Day 0	TW1	TW4	TW12	EOT	SVN12	Day 0	SVN12
1	nd	positive	neg	1	neg	neg	neg	nd	neg	neg	neg	neg
				2	neg	neg	neg	nd	neg	neg	neg	neg
2	nd	positive	neg	3	neg	neg	neg	nd	neg	neg	neg	reactive
				4	neg	neg	neg	nd	neg	neg	neg	reactive
3	2a	positive	>200.000	5	<15	neg	neg	neg	neg	neg	neg	reactive
				6	<15	neg	neg	neg	neg	neg	neg	reactive
4	2a	positive	>200.000	7	<15	neg	neg	neg	neg	neg	neg	reactive

HCV = Hepatitis C; GT = genotype; Ab = antibody; TW = therapy week; nd = not determined; EOT = end of treatment; SVN12 = Sustained virological negativity 12 weeks after end of treatment.

**Table 3 jcm-10-00089-t003:** Baseline characteristics of HCV positive donors and of HCV negative recipients.

	EOT	SVN12
Creatinine, mg/dl	1.3 (±0.4)	1.1 (±0.3)
Biopsy proven acute rejection	none	none
SAE	*n*	
Delayed graft function	3	
Renal biopsy	1	
Major bleeding	1	
Urosepsis	1	

**Table 4 jcm-10-00089-t004:** Long-term follow-up.

Patient No.	Creatinine mg/dL 2 Years after Transplantation	Albuminuria mg/g Creatinine in Spot Urine	Rejection/Biopsies	Unscheduled Hospitalizations
1	1.11	3	no/no	none
2	1.32	9	no/yes (showing ATN)	none
3	1.52	3	no/no	none
4	1.09	18	no/no	none
5	0.90	7	no/no	none
6	1.04	100	no/no	none
7	3.83	39	no/no	2 hospitalizations due to pseudomonas urosepsis Exitus letalis due to MSSA-Sepsis; 12 months post kidney Tx

## Data Availability

The data presented in this study are available on request from the corresponding author. The data are not publicly available due to General Data Protection Regulation.

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
