# Peer review of "Pan-Genotype Pre-Exposure Prophylaxis (PrEP) Allows Transplantation of HCV-Positive Donor Kidneys to Negative Transplant Recipients"

_jcm, 2020, doi:10.3390/jcm10010089_

Round 1

Reviewer 1 Report

Due to the worldwide shortage of organs this study on HCV positive kidney donors for seronegative recipients shows a relatively safe way to reduce waiting time particularely for patients who already have been waiting for many years.

Although the number of patients is quite small (n=7) the presented results demonstrate that the acceptance of HCV+ donors was safe and well tolerated under the protection of PrEP DAA in renal transplantation.

Furthermore it is important to see that no side effects could be observed in the long term f/u of that cohort.

How many organ donors are HCV positive?

It would be of interest to add the total number or percentage of HCV+ kidneys allocated via Eurotransplant per year.

Author Response

Reviewer 1:

  1. How many organ donors are HCV positive?
    It would be of interest to add the total number or percentage of HCV+ kidneys allocated via Eurotransplant per year.

Answer:
Thanks for this question on HCV positive donors. Recently, we asked Eurotransplant (ET) to give us new data on offered and utilized donors with HCV positive donor status.

We summarized the numbers of kidney transplantations from HCV-positive donors in ET area below. Between 2017 to 2019, out of n=65 HCV+ donors, n=128 HCV+ kidneys were offered, of which n=86 were transplanted over this period. Still, there is a 32% discard rate of HCV+ donor kidneys.  Interestingly, only n= 29 transplantations from HCV+ donors in contrast to n=3240 kidney transplantations from non-HCV donors took place on average per year. So far, the transplantation of HCV+ kidneys reflects a small proportion in the entire ET program. But still, we are convincede that HCV+ organs are underused and should be considered as a source of additional donor organs for our waitlisted patiens.

We included a corresponding section in the manuscript (on page 2, Introduction).

Additionally you will find in the attached file the table that summarize the precise numbers of kidney transplantations from HCVAb Pos Donors in ET between 2017 - 2019

Reviewer 2 Report

This manuscript by Duerr and colleagues describes the results of a prospective trial of pre-exposure prophylactic treatment with direct-acting antivirals in Hep C D+/R- kidney transplantation. They present data on 7 kidney transplant recipients who received a kidney transplant from Hep C+ donors. They utilized the pan-genotypic regimen of daclatasvir and sofosbuvir. All patients achieved SVN at 12 weeks following treatment.

This is an interesting report and adds to the breadth of knowledge and confidence in utilizing kidneys for transplantation from Hep C+ donors. There are a number of limitations of this manuscript.

  1. There are a number of recent important studies that are not referenced or discussed. I would recommend including them and discussing these within the introduction and discussion sections. a) Feld JJ. et al. Lancet Gastroenterology & Hepatology Volume 5, Issue 7, July 2020. b) Durand et al. Ann Intern Med 2020. c) Sise et al. JASN 2020.
  2. More description of patient demographics is required. Characteristics such as cause of ESKD, PRA, HLA mismatch, XM at time of transplant. Furthermore, follow up information such as proteinuria would be helpful.
  3. I am curious of the background on this unique study design. This is a very interesting aspect of this study and warrants further discussion. Included in this discussion are how this study design was decided on upon and the n value for each category. This is very similar to an adaptive trial design commonly used in cancer chemotherapy studies.
  4. Generally, SVR (sustained virological response) is used as an outcome of interest, instead of SVN.
  5. There are some abbreviations (eg ET) that are not defined.
  6. Were liver enzyme assessments captured at each follow-up times.
  7. What was the induction and maintenance immunosuppression used in all patients.

Author Response

Reviewer 2:

  1. There are a number of recent important studies that are not referenced or discussed.

I would recommend including them and discussing these within the introduction and discussion sections. a) Feld JJ. et al. Lancet Gastroenterology & Hepatology Volume 5, Issue 7, July 2020. b) Durand et al. Ann Intern Med 2020. c)  Sise et al. JASN 2020

 Answer:

Thank you for these valuable suggestion. We added all mentioned studies in the discussion section (yellow highlighted).

  1. More description of patient demographics is required. Characteristics such as cause of ESKD, PRA, HLA mismatch, XM at time of transplant. Furthermore, follow up information such as proteinuria would be helpful.

Answer:

Thank you for these valuable suggestion. We added information on reasons of ESKD, PRA, HLA mismatch as well as cross match in table 1 as well as the results section, p.4:

“All recipients received their first graft and had a standard immunological risk with a negative crossmatch and a cPRA <15% in class I and II antigens.”

 Further details for proteinuria was included in table 4 also.

  1. I am curious of the background on this unique study design. This is a very interesting aspect of this study and warrants further discussion. Included in this discussion are how this study design was decided on upon and the n value for each category. This is very similar to an adaptive trial design commonly used in cancer chemotherapy studies.

Answer:

Thanks for this question. Since the concept of accepting HCV positive kidneys was new to the ethics committee and initially judged by them to be questionable, we agreed on a very cautious approach.

We started with a small number of study participants to identify early signals of low efficacy or safety and ended up in n=5 subjects for the first phase.

The underlying idea of how much transmission was deemed acceptable was based on our clinical experience. We considered a safety limit for HCV transmission as less than 10% to be clinically acceptable. We designed all consecutive phases, with the option to terminate the entire study after each phase if too many cases with transmissions(>10%)  had taken place. We ended up with n=5 in phase I (with the option of addition n=5 patients, in case one would have been tested HCV positive), n=10 in phase II, and the remaining n=24 in phase III. This distribution was a pragmatic decision to start with a small number of patients and increase the numbers of participants over the next phases of the study.

We had no template from other studies that could have served as a basis when creating the study design.

Finally, we decided to use a staggered study design in which we only proceeded to the next phase of the study after the efficacy of the treatment had been proven.

Still, we are convinced this was a very safe study design, where we have implemented many safety stops in case the therapeutic efficacy would have failed. The staggered design details are summarized in figure 3 and in the method section, page 2 “study design.”

  1. Generally, SVR (sustained virological response) is used as an outcome of interest, instead of SVN.

Answer:

Thanks for this comment. We have carefully considered using the term sustained virological negativity (SVN) because our patients had never experienced any HCV infection. We are aware that the term sustained virological response (SVR) is widely used to describe therapeutic response in HCV trials. However, our trial included chronic HCV infected patients that have experienced with the virus and can respond to a specific DAA treatment. As said, our patients in the present study never had HCV, so they cannot respond with a virological response.

  1. There are some abbreviations (e.g. ET) that are not defined.

Answer:

We completed the abbreviations on table 1: ET=EuroTransplant and ICB= Intracranial bleeding and in figure 3: RTx=Renal Transplantation in the manuscript.

  1. Were liver enzyme assessments captured at each follow-up times.

Answer:

Liver enzymes were regualry monitored. No elevation was detected in patients 1-6 during the follow-up time. Patient 7 showed a mild elevation (<2-fold) of ALAT and ASAT during admission hospital admission with sepsis.

  1. What was the induction and maintenance immunosuppression used in all patients.

Answer:

Thank you for this question.  Immunosuppression was given according to our center policy (induction therapy with basiliximab, maintenance with tacrolimus, mycophenolate and steroids). The information is provided in the Materials and Methods section.

Round 2

Reviewer 2 Report

Thank you for the edits made to your manuscript. All questions were adequately dealt with.